# H-Dot Mediated Nanotherapeutics Mitigate Systemic Toxicity of Platinum-Based Anticancer Drugs

**DOI:** 10.3390/ijms242015466

**Published:** 2023-10-23

**Authors:** Atsushi Yamashita, Seung Hun Park, Lingxue Zeng, Wesley R. Stiles, Sung Ahn, Kai Bao, Jonghan Kim, Homan Kang, Hak Soo Choi

**Affiliations:** 1Gordon Center for Medical Imaging, Department of Radiology, Massachusetts General Hospital and Harvard Medical School, Boston, MA 02114, USA; ayamashita@mgh.harvard.edu (A.Y.); spark54@mgh.harvard.edu (S.H.P.); lingxue_zeng@student.uml.edu (L.Z.); wstiles1@mgh.harvard.edu (W.R.S.); sahn8@mgh.harvard.edu (S.A.); kbao@mgh.harvard.edu (K.B.); jonghan_kim@uml.edu (J.K.); hkang7@mgh.harvard.edu (H.K.); 2Department of Biomedical & Nutritional Sciences, University of Massachusetts Lowell, Lowell, MA 01854, USA

**Keywords:** biodistribution, cancer therapy, drug delivery, platinum-based drugs, rapid clearance

## Abstract

Platinum-based anticancer agents have revolutionized oncological treatments globally. However, their therapeutic efficacy is often accompanied by systemic toxicity. Carboplatin, recognized for its relatively lower toxicity profile than cisplatin, still presents off-target toxicities, including dose-dependent cardiotoxicity, neurotoxicity, and myelosuppression. In this study, we demonstrate a delivery strategy of carboplatin to mitigate its off-target toxicity by leveraging the potential of zwitterionic nanocarrier, H-dot. The designed carboplatin/H-dot complex (Car/H-dot) exhibits rapid drug release kinetics and notable accumulation in proximity to tumor sites, indicative of amplified tumor targeting precision. Intriguingly, the Car/H-dot shows remarkable efficacy in eliminating tumors across insulinoma animal models. Encouragingly, concerns linked to carboplatin-induced cardiotoxicity are effectively alleviated by adopting the Car/H-dot nanotherapeutic approach. This pioneering investigation not only underscores the viability of H-dot as an organic nanocarrier for platinum drugs but also emphasizes its pivotal role in ameliorating associated toxicities. Thus, this study heralds a promising advancement in refining the therapeutic landscape of platinum-based chemotherapy.

## 1. Introduction

The clinical introduction of platinum (Pt) compounds as anticancer agents marked a significant milestone in 1978; a diverse array of Pt anticancer drugs has emerged for cancer therapeutics [1]. Pt agents exert their chemotherapeutic effects by binding to nuclear DNA and generating reactive oxygen species (ROS), yet their nonspecific distribution leads to systemic toxicity [2,3,4]. Consequently, Pt drugs give rise to severe undesired effects, including dose-dependent toxicity, notably nephrotoxicity, neurotoxicity, ototoxicity, and myelosuppression [5], with prolonged administration causing extensive harm to normal tissues [6]. Notably, carboplatin, exhibiting lower toxicity than cisplatin, has been globally applied in various cancer treatments encompassing pancreatic, ovarian, head and neck, lung, and other cancers. Yet, its off-target toxicities, such as cardiotoxicity, neurotoxicity, and myelosuppression, remain challenging [7,8,9].

Given the substantial therapeutic impact of first-line Pt drugs on tumor tissues, diverse strategies have emerged to mitigate harm to normal tissues. These strategies encompass encapsulation within liposomes or polymeric micelles [10,11,12] and nanoparticle-based drug delivery [13,14]. These approaches augment the tumor targetability of encapsulated drugs, facilitated by the enhanced permeability and retention (EPR) effect. The EPR effect, acknowledged as a universal pathophysiological trait of solid tumors, fundamentally underpins the conceptualization and development of tumor-targeted delivery of anticancer drugs [15,16,17]. Previously, Kang et al. unveiled the intriguing phenomenon of the size-dependent EPR effect [18], where smaller polyethylene glycol (PEG) entities (≤20 kDa) exhibit a remarkable proclivity for tumor targeting while showing minimal nonspecific uptake [18]. Harvard-dots (H-dots) have been designed to deliver hydrophobic anticancer drugs, including imatinib, gefitinib, and genistein, with high delivery efficiency via the EPR effect relying on small-sized nanotherapeutics [19,20,21]. H-dots also offer reducing potential off-target toxicities of anticancer drugs through rapid renal clearance, of which the hydrodynamic diameter is smaller than the glomerular filtration threshold, approximately 5.5 nm [19]. Here, we chose carboplatin as the encapsulated Pt drug for H-dot complexation since it shows systemic toxicity including myelosuppression and cardiotoxicity with rare cases of nephrotoxicity, while cisplatin and oxaliplatin cause renal toxicity by damaging the kidneys [7,8,9]. In addition, the water solubility of carboplatin (14 mg/mL) may be beneficial to achieve quick release from the H-dot complexes before urinal excretion. We demonstrate the delivery of carboplatin using the EPR effect of H-dot by forming an inclusion complex to manifest enhanced therapeutic efficacy while mitigating the adverse effects through rapid renal clearance. 

## 2. Results

### 2.1. Preparation of Car/H-Dot Nanocomplex

H-dot was synthesized as previously reported [19]. Briefly, the positive charges of ε-poly-L-lysine (EPL) were converted to be zwitterionic by adding approximately 50% of carboxylates using succinic anhydrides. Then, to monitor its behavior in the body, ZW800-1C was conjugated on the terminal amine of EPL via the conventional NHS ester reaction [22,23]. Next, 8–10 β-cyclodextrin (β-CD) moieties were conjugated on the primary amines of EPL. Finally, carboplatin (Car; MW = 371.25 Da) was loaded into the hydrophobic cavity of β-CD (Figure 1a) by mixing the equimolar solutions and freeze-drying [20]. The loaded amount of Car in H-dots was determined by the UV absorbance increase at 230 nm, of which the complexation ratio was calculated to be 7.9 (Figure 1b and Appendix A). The water solubility of Car is approximately 37.7 mM (14 mg/mL), which was increased about 13-fold by adding H-dots in the solution (~0.5 M). The hydrodynamic diameter of H-dot and Car/H-dot were 6.56 and 6.53 nm, respectively (Appendix A), in which small size of drug carriers are suitable for tumor targeting by EPR effect as well as reducing off-target toxicity with rapid renal clearance [18,19,20,21]. The optical properties of the Car/H-dot nanocomplex were measured by UV-Vis-NIR spectroscopy and near-infrared (NIR) fluorescence imaging system (Figure 1c). The maximum wavelengths of the fluorescence emission peaks for Car/H-dot were 795 nm.

### 2.2. Drug Release Test

Before assessing in vitro antitumor efficacy, the drug release efficacy of Car/H-dot at the tumor microenvironment was tested in phosphate-buffered saline (PBS, pH 7.4) at 37 °C for up to 24 h. The released amount of Car from the inclusion complexes was determined by measuring the changes in UV absorption at 230 nm. Approximately 90% and 99% of Car was dissociated from the Car/H-dot nanocomplex 8 h and 24 h post-incubation, respectively (Figure 1d). Since the small-size EPR effect and rapid clearance of H-dot have been well demonstrated previously [18,19,20,21], the rapid release of Car from the H-dot nanocomplex is advantageous to the in vivo therapeutic efficacy of tumor and potentially minimizes the adverse effects of Car. 

### 2.3. In Vitro Therapeutic Efficacy and Cytotoxicity Evaluation

The therapeutic efficacy of Car/H-dot nanocomplexes was evaluated in the NIT-1 insulinoma cell line and compared with NIH3T3 and HEK293 normal cell lines by using CCK-8 assay and microscopic observation (Figure 2). The Car or Car/H-dot treated group clearly showed dose-dependent cell deaths at 24 h post-treatment. The Car or Car/H-dot treated group showed almost the same toxicity profiles, half-maximal response concentration (EC50) values, and similar microscopic images at the same concentration conditions (Figure 2a and Appendix A). The toxicity profiles and EC_50_ values of Car/H-dot nanocomplexes are obtained in vitro toxicity assay 24 h post-treatment based on the release pattern of nanocomplexes (Figure 1d).

There was no significant evidence of cytotoxicity or cell morphological changes as compared with the no-treatment control group (H-dot without drugs), even at the highest dose (5000 µM; Appendix A). These results indicate that the Car release from H-dot complexes strongly correlated with in vitro antitumor activity. Similar results were obtained from the toxicity assays with other cell lines. NIH3T3 fibroblast (Figure 2b and Appendix A) and HEK293 kidney (Figure 2c and Appendix A) cell lines derived from the embryo were also assigned for in vitro cytotoxicity assay to compare the sensitivity against Car and Car/H-dot. Interestingly, the sensitivity of Car for NIT-1 insulinoma cell lines was higher (lower EC_50_ values ≈ 80 µM) than that on NIH3T3 and HEK293 normal cell lines, indicating that the Car could inhibit the proliferation of NIT-1 insulinoma cells more specifically regardless of free drugs or drug/H-dot complexes (Figure 2d). In the case of NIH3T3 cells, the EC_50_ value of Car and Car/H-dot were 332.9 mM and 374.7 mM, respectively. Similarly, 774.2 mM and 931.9 mM of EC_50_ values were obtained in HEK293 cells treated with Car alone and Car/H-dot groups, respectively. Similar results are reported on NIH3T3 and breast cancer cell lines (MCF-7 and MDA-MB-231) [24]. Since malignant cells are highly generating reactive oxygen species (ROS) [24,25,26,27], they are more sensitive against generated ROS, including chemotherapy with Pt drugs [4,26,28,29]. We confirm that H-dots release drugs efficiently in the cells without causing additional cytotoxicity or cell morphological changes at high concentrations (>5000 µM).

### 2.4. In Vivo Biodistribution and Pharmacokinetics

Car/H-dot nanocomplex was evaluated to improve the tumor accumulation/therapeutic efficacy of Car using insulinoma-bearing mice and to reduce the adverse effects of Car in major organs. Firstly, the biodistribution of H/dot and Car/H-dot were investigated in CD-1 mice 4 h before imaging (Figure 3).

The signal-to-background ratio (SBR) in the heart, lung, liver, pancreas, spleen, kidney, duodenum, intestine, and muscle in the mice treated with H-dot alone and Car/H-dot were similar, indicating that the encapsulation of Car has little to no impact on the organ distribution of H-dots (Figure 3a–c). Then, the pharmacokinetics of H-dot and Car/H-dot were examined in CD-1 mice after a single intravenous injection (Figure 3d,e). The NIR fluorescent signal intensity from mouse serum collected at predetermined time points was measured to obtain the blood concentration decay curves of H-dot and Car/H-dot nanocomplex. The results demonstrated that both H-dot and Car/H-dot complexes exhibited pharmacokinetic behaviors consistent with the two-compartment model. H-dots distributed rapidly into major organs (*t*_½α_ = 4.27 ± 0.61 and 4.48 ± 4.45 min for H-dot and Car/H-dot, respectively) and then eliminated quickly from the body (*t*_½β_ = 46.72 ± 4.59 and 43.43 ± 27.43 min for H-dot and Car/H-dot, respectively) with a fast plasma clearance rate (0.135 mL/min and 0.128 mL/min for H-dot and Car/H-dot, respectively). The values for the volume of distribution (250.3 mL/kg for H-dot and 216.4 mL/kg for Car/H-dot) are close to the volume of the extracellular fluids (200 mL/kg) indicating that the two complexes distributed into the whole body without specifically projecting to the peripheral compartment. Indeed, more than 77% of H-dots were eventually excreted through urine within 4 h. These results suggest that Car/H-dots experience non-toxic events in the body and are conducive to effective renal clearance. Overall, the encapsulation of Car did not change the biodistribution and pharmacokinetic characteristics of H-dots. These results are not surprising because our previous reports showed the same trend with encapsulation of imatinib as well as gefitinib and genistein for the treatment of lung and gastrointestinal stromal tumors (GIST) [19,21].

### 2.5. In Vivo Tumor Targeting and Therapeutic Efficacy of the Car/H-Dot

Encouraged by the superb results of in vitro drug release and toxicity assay, in vivo tumor targetability and tumor inhibition efficacy of those Car/H-dot complexes were evaluated in insulinoma mice (Figure 4). Insulinoma-bearing NOD/ShiLt-Tg(RipTAg)1Lt/J mice, purchased from Jackson Laboratories (Bar Harbor, ME) are transgenic mice that spontaneously induce pancreatic neuroendocrine tumors (panNETs) after 10 weeks old [30]. To avoid harsh conditions for insulinoma mice, the treatment dose of Car and Car/H-dot was decided as 30 mg/kg of drug (185 mg/kg of Car/H-dot) to be a quarter of the medium lethal dose (LD_50_) [31], Figure 4a shows color and NIR images of the resected spleen and pancreas with tumors from insulinoma mice on 2 d post-intraperitoneal injection. Strong fluorescence signals of Car/H-dot were observed within the tumor mass (tumor interstitial fluid), similar to our previous results [32]. Presumably, the EPR effect arrowed for quick accumulation and retention of H-dot complexes, resulting in a higher fluorescence signal around the tumor surrounding area, which was retained more than 2 d post-injection. Indeed, active vascularization around panNET insulinoma has been well-reported in humans and mice [33,34,35]. The resected tumors from the mouse injected with Car and Car/H-dot were examined by ICP-MS measurement to confirm the penetration and distribution of Pt drugs (Figure 4b). Indeed, accumulation of Car was detected after a single administration of both Car and Car/H-dot. This result suggests that Car/H-dots complexes could deliver drugs deep into tumor tissues due to the efficient release from Car/H-dot around the tumors. These results strongly supported our hypothesis that early release from the Car/H-dot complexes is a suitable strategy for the therapeutic efficacy of Car. 

Next, in vivo, the antitumor efficacy of Car/H-dot nanocomplexes was evaluated in insulinoma-bearing mice. In this study, we injected 30 mg/kg of Car and 185 mg/kg of Car/H-dot, including the same dose of Car, into insulinoma mice (12–14 weeks old). The treatment was repeated 5 times within 12 d, and the body weight of treated mice was monitored every day. The number and weight of tumors were measured after finishing treatments on day 21 to confirm the treatment efficacy of Car and Car/H-dot (Figure 4c,d). As we expected, the Car/H-dot group achieved excellent treatment of insulinoma, which eliminated the panNET insulinoma nodule completely. Similarly, the injection of Car alone also reduced insulinoma tumors’ number and weight, while several tumor nodes remained. These results were consistent with the in vitro toxicity assay (Figure 2), which showed similar therapeutic efficacy between Car alone and Car/H-dot nanocomplex. In the case of in vivo situations, the quick accumulation of Car/H-dot around the tumor area might assist in eliminating panNET insulinoma. Treatment of panNET using cisplatin has been approved by the US Food and Drug Administration (FDA), but not yet treatment with Car. However, in this study, the Car and Car/H-dot group showed sufficient treatment of panNET. These results strongly suggested that our H-dot complex system can push the limit of therapeutic strategy for panNET by Car, which reduces the adverse effects compared with Cis.

### 2.6. Systemic Toxicity Evaluation of Car and Car/H-Dot Complex

During the experiment period, the body weight of the treated mice was monitored every day (Figure 4e). After sacrificing the mice on day 21, the systemic toxicity of Car alone and Car/H-dot complex were further evaluated by organ weight, histopathological examinations, and biochemical analysis. The body weight of control insulinoma mice slowly decreased until the end of the monitoring due to the progression of insulinoma. On the other hand, Car and Car/H-dot groups did not show body weight loss during the experiment periods due to the treatment effect of insulinoma. Indeed, a significant difference in treatment effects was observed in both Car and Car/H-dot treated groups (* *p* < 0.05; Figure 4c,d). Several adverse effects of Car have been reported, such as bone marrow suppression [36], ototoxicity [37], and cardiotoxicity [38,39,40] in both single and combination therapy with other drugs. In particular, the cardiotoxicity of Car proved to be fatal, causing apoptotic cell deaths and loss of cardiac fibril, significantly reducing mice survival rate [38].

Similarly, the cardiac myofibril clearly disappeared from the heart treated with Car, while relatively clear fibers were observed in the hearts of Car/H-dot and control groups (Figure 5a). The rest of the major organs, including the lung, liver, kidney, and spleen, show nonsignificant toxic signs (Figure 5b and Appendix A). Indeed, ALT, BUN, and creatinine levels in plasma samples from the treated mice were similar to those in the control group (Figure 5b,c). In addition, the Car/H-dot complex did not induce significant renal impairment despite their exclusive renal clearance. These are not surprising because the less adverse effects of Car compared with other Pt drugs have been well reported in terms of hepatotoxicity and nephrotoxicity [41,42]. Even though more than three times higher dose with a single administration of Car (90 mg/kg) and Car/H-dot (555 mg/kg), significant toxic signs were not detected at post-1-week administration (Appendix A). Interestingly, the AST level in the Car/H-dot group was significantly reduced compared with the non-treated groups at day 21 (Figure 5b). It has been reported that the insulinoma mice model frequently causes metastasis to the liver [33,43], resulting in a reduction of liver function, and increasing the AST and ALT levels in the blood. Therefore, Car/H-dot might treat micrometastasis in the liver. In addition, Car/H-dot seems to prevent cardiotoxicity of Car, which also might contribute to reducing the AST level due to the prevention of cardiac apoptosis because cardiac cell death increases blood AST levels [44]. While the insulinoma treatment effects of Car were similar to Car/H-dot, the AST level was not reduced in the Car group, presumably due to the cardiotoxicity of Car. Overall, Car/H-dot showed little to no toxic signs, even improving the cardiac fibril and blood AST levels. These results clearly indicate that Car/H-dot nanocomplex prevents adverse effects of Car without reduction of treatment effect on panNET insulinoma. Since tumor treatment with Car/H-dot shows comparable toxicity with improved renal clearance compared with Car, this combination can be a more promising strategy for reducing side effects with sufficient therapy of tumors. 

## 3. Discussion

The treatment of panNET and avoiding normal pancreas represent a significant unmet clinical need. Intraoperative visual inspection and palpation performed by the surgeon is considered the most reliable source for panNET detection, which often results in incomplete resection of the tumor, with a 5-year disease recurrence rate of 10% and 97% of patients without and with liver metastasis, respectively [45,46,47,48]. Thus, complete tumor treatment is the only option for improving patient survival [48,49]. However, we previously demonstrated the difficulty of delivering drug molecules (>300 Da) into panNET tissue due to the high tumor interstitial pressure [32]. Therefore, we decided to design a new delivery system for a quick-release of small-molecule anticancer drugs to achieve sufficient treatment of panNET with reduced adverse effects. Many drug carriers have been widely accepted for various kinds of tumor treatment for delivering the drugs into tumor tissue [10,11,12,13,14]. 

We hypothesized that H-dot drug delivery systems might be able to achieve efficient delivery of chemotherapeutics with fewer adverse effects mainly due to the reduced off-target delivery and rapid renal clearance. Indeed, the drug/H-dot system showed high efficacy in delivering gefitinib and genistein to lung cancers [20] and imatinib to gastrointestinal tumors [19,21]. In this study, we successfully developed a quick-release system of Car from H-dot due to the hydrophilic properties of the drug. Over 90% of encapsulated Car was released from the Car/H-dot nanocomplex 8 h post-incubation. Since one of the novel functions of the H-dot system is rapid renal clearance to reduce off-target effects of encapsulated drugs and carrier itself, the quick release of Car before urinary excretion would be critical for panNET treatment. Indeed, sufficient treatment of panNET was observed both in vitro and in vivo, which did not disturb the efficacy of Car alone but rather slightly enhanced the in vivo treatment effects. The outstanding finding is that a quick drug release system of Car/H-dot could eliminate the panNET completely without drug-carriers penetration into the tumor tissue. While the limited penetration of drug delivery systems into deep tumor tissue has been one of the major limitations of nanoplatforms in treating panNET insulinoma [32], efficient Car release from the Car/H-dot complex around the tumor area was beneficial to the sufficient treatment. Furthermore, the H-dot delivery system prevented the cardiotoxicity of Car by encapsulation in the cavity of β-CD, which reduced cellular uptake (Figure 2) and nonspecific uptake in organs (Figure 5). Our results demonstrated that Car/H-dot could eliminate panNET with fewer adverse effects. However, further investigations are required to clarify the mechanisms.

## 4. Materials and Methods

### 4.1. Reagents and Materials

Epsilon-poly-L-lysine (ε-poly-L-lysine, EPL; MW ~3900) was purchased from BOC Sciences (Shirley, NY, USA). β-cyclodextrin (β-CD; MW ~1134) was purchased from Tokyo Chemical Industry (Tokyo, Japan). Acetic acid, ethyl acetate (EA), Dess-Martin periodinane (DMP), ninhydrin, succinic anhydride, deuterium oxide (D_2_O), anhydrous dimethyl sulfoxide (DMSO), sodium acetate, sodium borohydride, sodium hydroxide, bovine serum albumin (BSA), aspartate aminotransferase (AST) assay kit, and alanine aminotransferase (ALT) assay kit were purchased from Millipore Sigma (Burlington, MA, USA). Ninhydrin agent and succinic anhydride (SA) were purchased from Acros Organics (Morris Plains, NJ, USA). Acetone was purchased from Fisher Scientific (Pittsburgh, PA, USA). Carboplatin (Car) was purchased from Selleck Chemicals (Houston, TX, USA). Dulbecco’s modified eagle medium (DMEM), fetal bovine serum (FBS), and penicillin/streptomycin solution were purchased from Thermo Fisher Scientific (Waltham, MA, USA). The NIT-1, NIH3T3, and HEK293 cells were purchased from the American Type Culture Collection (Rockville, MD, USA). Cell Counting Kit-8 (CCK-8) was purchased from Dojindo Molecular Technologies (Kumamoto, Japan). The biochemical assay reagent kits for serum creatinine and urea assay were purchased from Cayman Chemical (Ann Arbor, MI, USA) and Bioassay Systems (Hayward, CA, USA), respectively.

### 4.2. Synthesis of H-Dot

H-dot was synthesized according to previously reported methods. To prepare mono-aldehyde β-CD (Ald-CD), β-CD was oxidized with DMP in anhydrous DMSO at room temperature for 2 h, followed by recrystallized in excess EA/acetone (20% *v/v*). The supernatant was discarded, and the precipitate was lyophilized to obtain the Ald-CD. The obtained Ald-CD powder and EPL were dissolved in pH-adjusted phosphate-buffered saline (PBS, pH 8.0) and stirred for 2 h. To reduce the Schiff base to a secondary amine, sodium borohydride was added, and the mixture was stirred for an additional 72 h at room temperature. Dynamic dialysis was carried out in a cellulose membrane with a molecular weight cutoff (MWCO) of 6–8 kDa for 24 h against DIW. The dialyzed solution was frozen at −80 °C and lyophilized to retrieve an off-white fluffy solid. To characterize the β-CD-conjugated EPL (CDPL), the number of β-CD grafted to EPL was determined using 1 H-NMR spectroscopy. 

To conjugate an NIR dye, ZW800-1C-NHS in DMSO was added dropwise into the CDPL solution (PBS, pH8.0) and stirred for 3 h with maintained pH of 8.0 by adding 0.6 M aqueous sodium hydroxide as needed. The mixture was precipitated in excess EA/acetone (20% *v/v*) and centrifuged at 3000 rpm for 15 min. The supernatant was discarded, and the dissolution of ZW800-CDPL in DIW and precipitation in EA/acetone was repeated two more times. The resulting product was dried in vacuo overnight to yield a green solid ZW800-CDPL. To obtain zwitterionic H-dots, succinic anhydride solution in DMSO was added to a solution of ZW800-CDPL in PBS (pH 8.0), and the mixture was stirred for 30 min at room temperature. The pH was maintained at 8.0 throughout the reaction by adding 0.6 M sodium hydroxide as needed. Once the ninhydrin test confirmed the partial succinylation, the product was dialyzed in a cellulose membrane with a molecular weight cutoff (MWCO) of 6~8 kDa for 24 h against DIW. The resulting solution was frozen at −80 °C and lyophilized to afford the final zwitterionic NIR fluorophore-CDPL±, also known as H-dot.

### 4.3. Preparation of Car/H-Dot Inclusion Complexes

To prepare the Car/H-dot complex, Car powder was dissolved in DI water (2 mM). H-dot powder was dissolved in DI water (1 mM). Afterward, the above solutions were mixed at a volume ratio of 1:1 and vortexed for two hours at room temperature. Then, the solution was centrifuged at 14,000 rcf for 10 min to precipitate impurities and obtain the product complex from the supernatant. The collected supernatant was lyophilized to obtain the Car/H-dot powder. The H-dot to Car molar ratio was determined using absorbance measurement (230 nm).

### 4.4. Measurement of Hydrodynamic Diameter of H-Dots

The size of H-dots was measured using size-exclusion chromatography (SEC) analysis on the Agilent HPLC system consisting of a 1260 binary pump with a 1260 ALS injector, a 35900E Photodiode Array detector (Agilent, Santa Clara, CA, USA, 200–800 nm), and a 2475 multi-wavelength fluorescence detector (Waters, Ex 770 nm and Em 790 nm). A portion of the eluent flowed into the PDA equipped with an Ultrahydrogel 2000 (7.8 × 300 mm) SEC column. The mobile phase was 0.1% formic acid in water for 30 min with a flow rate of 0.75 mL min^−1^. The hydrodynamic diameter of H-dots was obtained from the corresponding hydrodynamic diameter of standard proteins: Aprotinin, 2.0 nm; ribonuclease, 3.3 nm; myoglobin, 4.1 nm; ovalbumin, 6.1 nm; bovine serum albumin (BSA), 7.8 nm; immunoglobulin G (IgG), 10.6 nm; thyroglobulin, 17.0 nm.

### 4.5. Drug Release Tests of Car/H-Dot Complexes

The drug release tests of Car/H-dot complexes were carried out using rapid equilibrium dialysis devices (8 kDa MWCO, Thermo Fisher Scientific, Waltham, MA, USA). Car/H-dot complexes were separately dissolved in PBS solutions at a concentration of 500 μM. The sample chambers were filled with 200 μL of complex solution, and corresponding buffer chambers were filled with 400 μL of PBS solution. The plate was put on an up-and-down shaker at 20 rpm at 37 °C. 200 μL of the release solutions were pipetted from the buffer chambers at each sampling time: 0.25, 0.5, 1, 1.5, 2, 4, 6, 8, and 24 h, and 200 μL of fresh PBS was added. The concentration of the Car was calculated by measuring UV absorbances at 230 nm. The following equation calculated the accumulated drug release percentage (*Q_n_*):(1)$$Q_{n}=\left[\left(\sum_{i=1}^{n-1}C_{i}*0.2+C_{n}*0.4\right)/\left(C_{n}*0.2\right)\right]*100\%\;\left(1\leq n\leq 9\right)$$

*C_i_* and *C_n_* are the drug concentrations in the buffer chamber at each time point, and *C*_0_ is the concentration of the Car in the sample.

### 4.6. In Vitro Therapeutic Efficacy Test of Car/H-Dot Complexes

NIT-1, NIH3T3, and HEK293 cells were incubated in DMEM supplemented with 10% FBS and 1% penicillin/streptomycin in a humidified incubator at 37 °C before experiments. For the in vitro efficacy test, cells were seeded at a density of 4 × 10^3^ cells/well in 96-well plates. The plates were incubated for 24 h before treatment with H-dot, Car, and Car/H-dot ranging from 0.1 to 5000 μM. The cells without any treatments were used as a control. The therapeutic efficacy of each drug was evaluated by the CCK-8 (cell counting kit-8) assay and microscopy. For the CCK-8 assay,10 µL of CCK-8 solution was added to each well in the plates with an incubation time of 4 h, and then the absorbance was measured at 450 nm using a Cytation 5 (Bio Tek, Winooski, VT, USA ), which is equipped with a plate reader. All experiments were carried out with three replicates. The survival rate was calculated according to the equation below:Survival rate (%) = [(*A*_sample_ − *A*_b_)/(*A*_c_ − *A*_b_)] × 100(2)
where *A*_sample_, *A*_b_, and *A*_c_ are absorption values from drug treatments, blank, and no treatment control, respectively. The following equation calculated the mean EC50 value:*E* = Bottom + [(Top − Bottom)/1 + (*x*/*EC*_50_)^−Hill coefficient^](3)
where *E* is the % cell viability, the Top and Bottom are plateaus in the % cell viability, *x* is the logarithm of the concentration, and the Hill coefficient reflects the slope of the curve.

### 4.7. In Vivo Biodistribution and Pharmacokinetics of H-Dot and Car/H-Dot

Animals were housed in an AAALAC-certified facility and studied under the supervision of MGH IACUC per the approved institutional protocol (2016N000136). Before injection of treatments, six-week-old CD-1 mice (male; 25–30 g from Charles River Laboratories (Wilmington, MA, USA) were anesthetized with isoflurane and oxygen, and blood was sampled in capillary tubes (Fisher Scientific, Pittsburgh, PA, USA) at time point 0 min by slightly cutting the end of the tail. H-dot and Car/H-dot in saline were intravenously injected at the same dose level as the imaging experiments. Blood samples were obtained at 1, 3, 5, 10, 30, 60, 120, 180, and 240 min post-injection. The fluorescence intensities of serum samples in capillary tubes were measured to calculate distribution (*t*_1/2α_) and elimination (*t*_1/2β_) half-life values (*n* = 3 for each group). After 4 h post-injection, mice were sacrificed to image biodistribution and resected organs (liver, lung, spleen, kidney, stomach, brain, intestine, and bladder). The fluorescence and background intensities of a region of interest over each tissue were quantified using customized imaging software and ImageJ v1.48 (National Institutes of Health, Bethesda, MD, USA). The signal-to-background ratio (SBR) was calculated as SBR = fluorescence/background. Results were presented as a bi-exponential decay curve using Prism software version 9.0 (GraphPad, San Diego, CA, USA). Similarly, insulinoma-bearing NOD/ShiLt-Tg(RipTAg)1 Lt/J mice (Jackson Laboratories, Bar Harbor, ME, USA) were used for the evaluation of tumor targetability of Car/H-dot. The 4-week-old insulinoma mice (both male and female) were purchased and were maintained on a high glucose diet (8.4 g sugar in 250 mL water) until they were ready (12 weeks old) for the insulinoma treatment study. After 48 h injection of Car/H-dot (185 mg/kg), organs were resected for evaluation of tumor targetability by imaging analysis and measurement of Car amount. The amounts of Car in organs and tumors were analyzed by inductively coupled plasma mass spectroscopy (ICP-MS).

### 4.8. In Vivo Antitumor Efficacy Test

The insulinoma mice were divided into three treatment groups (*n* = 3–4) between 12 and 14 weeks old. Each group received different treatments: saline, 30 mg/kg Carboplatin, and 185 mg/kg Car/H-dot, given every 3 days 5 times within the 12-day. All reagents in saline were injected intraperitoneally. Mice were sacrificed on day 21 for ex vivo imaging and histological evaluations. Body weight was measured every day during the treatment and post-administration. The tumors were excised, and the number and weight were measured. 

### 4.9. Tissue Histopathology Evaluation

To determine the tissue distribution of the Car, Heart, liver, spleen, lung, and kidney tissues were removed from treated mice on day 21. The dissected tissues were trimmed and embedded in the Tissue-Tek optimum cutting temperature (OCT) compound (Saku Finetek, Torrance, CA, USA) without a pre-fixation step, and the tissue block was frozen at −80 °C. Ten-µm thick frozen sections were cut by a cryostat (Leica Biosystems, Deer Park, IL, USA). Then, those sections were stained with hematoxylin and eosin (H&E) for pathology observation under optical microscopy.

### 4.10. Toxicity Study

To evaluate the potential toxicity, blood samples were obtained by cardiac puncture at the end of the antitumor efficacy test (day 21). These blood samples were stored without anticoagulant and then centrifuged at 1500 rpm for 15 min at 4 °C. The serum was stored at −80 °C until further assays. The biochemical parameters tested were a liver function panel (aspartate aminotransferase (AST) and alanine aminotransferase (ALT)), and kidney function indication (blood urea nitrogen (BUN) and creatinine (CREA)). All parameters were tested using commercially available assay kits, and the absorbance was immediately measured by a plate reader.

### 4.11. Statistical Analysis

The fluorescence and background intensities of a region of interest over each tissue were quantified using customized imaging software and ImageJ v1.48 (National Institutes of Health, Bethesda, MD, USA). The signal-to-background ratio (SBR) was calculated as SBR = fluorescence/background, where the background is the fluorescence intensity of the muscle. Data are reported as mean ± s.e.m. with a minimum of three biological replicates. Student’s *t*-test statistical analysis was performed to evaluate the significance of the experimental data. A p-value of less than 0.05 was considered significant. The data was indicated with * *p* < 0.05, ** *p* < 0.01, and *** *p* < 0.001.

## 5. Conclusions

We confirmed that Car/H-dot nanocomplexes could target tumor surrounding areas by the EPR effect and showed sufficient treatment effect of Car/H-dot for in vitro and in vivo orthotopic panNET models. As almost 100% of Car release from the H-dot complex was observed within 24 h, the efficient release profile of Car/H-dot achieved the sufficient therapeutic effect of Car, which was accompanied by quick tumor targeting. In addition, Car/H-dot could prevent the loss of cardiac myofibrils and improve the blood AST level. Importantly, Car/H-dot nanocomplexes avoid the adverse effects of Car without reduction of the treatment efficacy for insulinoma. Thus, the complex formation between Car and H-dot showed beneficial functions for panNET insulinoma treatment. This critical achievement will help further optimize cancer treatment by H-dot complexes with several cancer therapeutics.

## Figures and Tables

**Figure 1 ijms-24-15466-f001:**
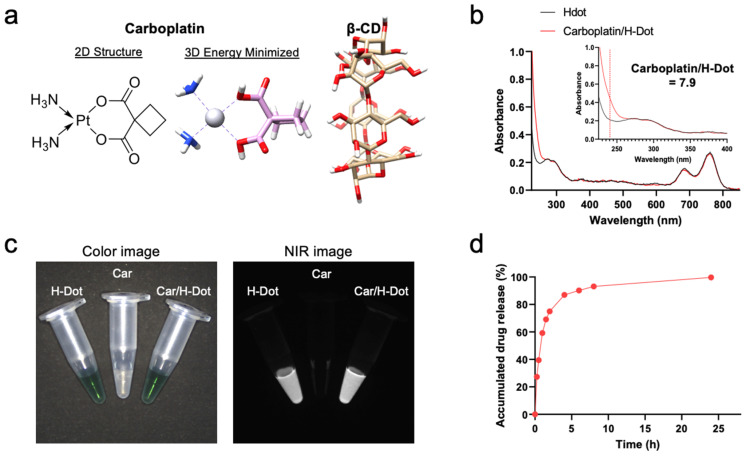
Nanocomplex formation and dissociation of carboplatin with H-dot: (**a**) Chemical structures of carboplatin (Car) and β-cyclodextrin (β-CD). (**b**) UV absorption spectra of H-dot (black line) and Car/H-Dot (red line). (**c**) NIR fluorescence images (800 nm channels) of H-dot and Car/H-dot nanocomplexes. (**d**) Cumulative drug release percentage of Car from H-dot nanocomplexes in PBS, pH 7.4 at 37 °C.

**Figure 2 ijms-24-15466-f002:**
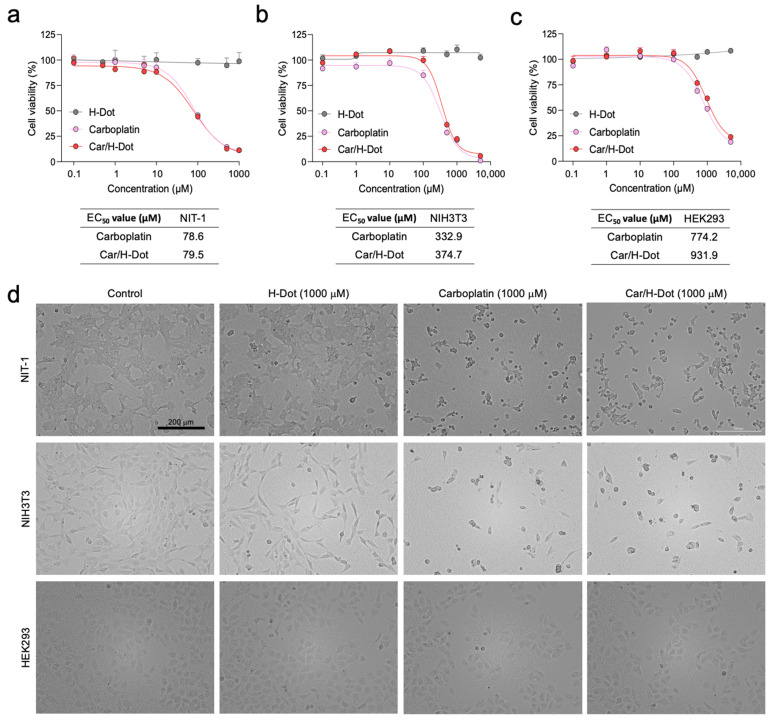
In vitro therapeutic efficacy test of Car and Car/H-dot nanocomplexes: Growth inhibitory effects of (**a**) NIT-1, (**b**) NIH3T3, and (**c**) HEK293 cells treated with different concentrations of empty H-dot, Car, and Car/H-dot for 24 h (*n* = 3, mean ± s.e.m.). (**d**) Microscopic cell images (×10) to compare the morphology and density of NIT-1, NIH3T3, and HEK293 cells after treatment with H-dot, Car, and Car/H-dot for 24 h. Scale bar: 200 µm.

**Figure 3 ijms-24-15466-f003:**
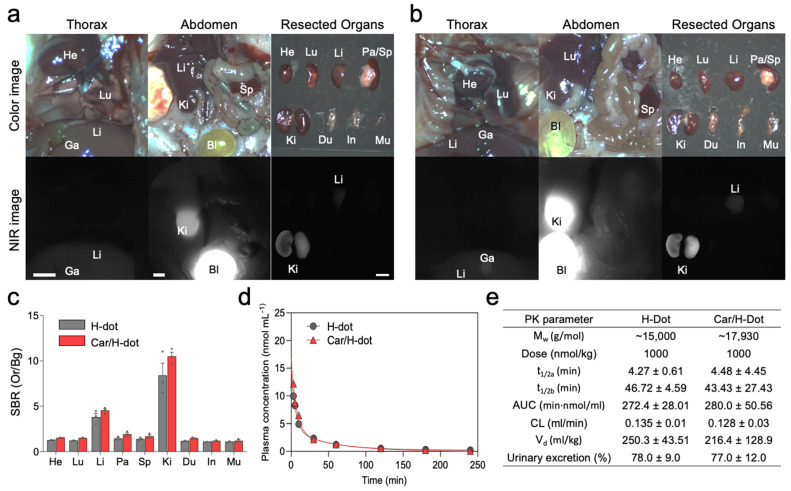
Biodistribution and pharmacokinetics of Car/H-dot nanocomplexes: H-dot (**a**) and Car/H-dot (**b**) were injected intravenously into CD-1 mice 4 h before imaging, and their NIR fluorescence images were recorded intraoperatively. Abbreviations used are Bl, bladder; Du, duodenum; Ga, gallbladder; He, heart; In, intestine; Ki, kidneys; Li, liver; Lu, lungs; Mu, muscle; Pa, pancreas; Sp, spleen. Scale bar: 5 mm. (**c**) Signal to background ratio (SBR) of each resected organ from mice injected with H-dot and Car/H-dot. (**d**) Plasma concentration curves of H-dot (black line) and Car/H-dot (red line) and their (**e**) pharmacokinetic parameters were calculated (*n* = 3 per group, mean ± s.e.m.).

**Figure 4 ijms-24-15466-f004:**
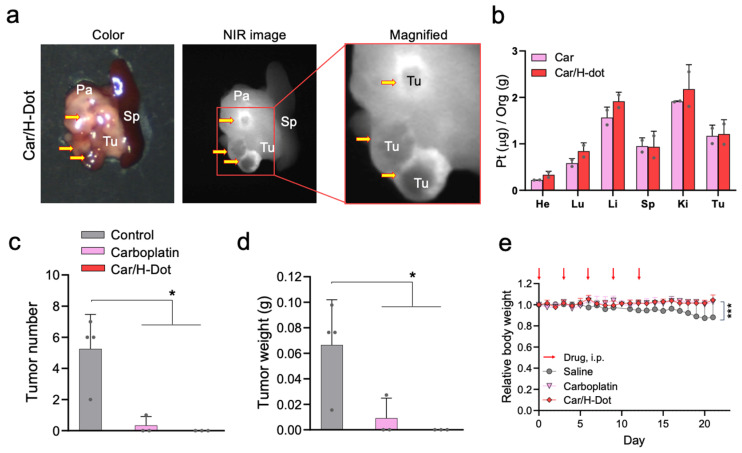
Tumor targeting and treatment effects of Car and Car/H-dot nanocomplex in insulinoma mice: (**a**) Color and NIR fluorescence images of the resected tumors treated with Car/H-dot 48 h post-intraperitoneal injection. Abbreviations used are Pa, pancreas; Sp, spleen; Tu, tumor (yellow arrows). (**b**) Quantitative analyses of Car and Car/H-dot amounts in resected organs measured by ICP-MS 48 h post-intraperitoneal injection. (*n* = 2 per group, mean ± s.e.m.). (**c**) Quantitative analyses of tumor numbers. No tumor was detected from Car/H-dot treated groups. (**d**) and tumor weights. No tumor was detected from Car/H-dot treated groups. (**e**) resected from insulinoma mice treated with saline, Car, and Car/H-dot (*n* = 4 per group, mean ± s.e.m.). (**e**) Changes in mice’s body weights during the treatments with Car and Car/H-dot (*n* = 3–4 per each group). * *p* < 0.05 and *** *p* < 0.001.

**Figure 5 ijms-24-15466-f005:**
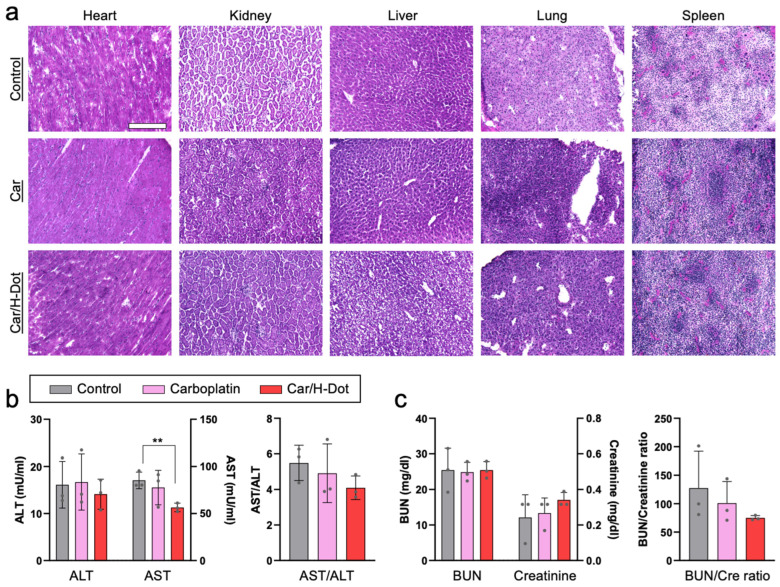
In vivo toxicity test of insulinoma mice treated with Car and Car/H-dot nanocomplex: (**a**) H&E staining images (20×) of the heart, liver, spleen, lung, and kidney in each treatment group. Scale bar: 200 µm. (**b**) Serum aspartate transferase (AST), alanine transferase (ALT), and AST/ALT ratio. (**c**) Blood urea nitrogen (BUN), creatinine (Cre), and BUN/Cre ratio. (*n* = 3 per each group). ** *p* < 0.01.

## Data Availability

All data are available in the main text or Appendix A.

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
