# Peer review of "H-Dot Mediated Nanotherapeutics Mitigate Systemic Toxicity of Platinum-Based Anticancer Drugs"

_ijms, 2023, doi:10.3390/ijms242015466_

Round 1

Reviewer 1 Report

Its a very well written manuscript, I appreciate the authors for carrying out an extensive scientific work. I have the following questions/comments: 

1.       Page 2, Line 72, it should be renal, not rnal

2.       Cells in our body are surrounded by a thick ECM, not many materials pass through these thick matrices, why do authors expect H-dots to penetrate deeply in tumour tissue?

3.       The authors suggest that encapsulating carboplatin reduces its cardiotoxicity, how? What is the hypothesis behind this, as long as the H-dots are not being targeted to a particular cell type using specific ligand-receptors, they can escape in the circulation.  

4.       What was the reason to choose, carboplatin for this specific study? There are other safer drugs available which can be made more effective using combination strategies.

5.       In figure 19(c), if possible, the authors should use a white background.

6.       Why was the PBS in drug release study, set at pH 6.0 and not 7.2?

7.       How is a early release of car from the nanocomplexes different from direct intake of car? Why was not a sustainable release over a period of time favourable for car delivery?

8.       Did the authors perform in vitro cytotoxicity at 48 hours, 72 hours? What was the difference observed?

Author Response

Its a very well written manuscript, I appreciate the authors for carrying out an extensive scientific work. I have the following questions/comments:

  1. Page 2, Line 72, it should be renal, not rnal.

We fixed the type accordingly.

  1. Cells in our body are surrounded by a thick ECM, not many materials pass through these thick matrices, why do authors expect H-dots to penetrate deeply in tumour tissue?

This is very important question. We previously reported that our H-dot complexes can penetrate tumors due to the complexed drugs: e.g., gefitinib and genistein for lung cancers [20] and imatinib for gastrointestinal stromal tumors [21]. The role of H-dot is to reduce nonspecific background uptake in the surrounding tissues. Car/H-dot, however, accumulates in the tumor interstitial space not in the tumors since the majority of panNETs are non-functional and do not express cell-type specific hormones [32]. In addition, molecules larger than 300 Da have poor targetability of the panNET tissue [32].

  1. Yin, X.; Cui, Y.; Kim, R. S.; Stiles, W. R.; Park, S. H.; Wang, H.; Ma, L.; Chen, L.; Baek, Y.; Kashiwagi, S.; Bao, K.; Ulumben, A.; Fukuda, T.; Kang, H.; Choi, H. S., Image-guided drug delivery of nanotheranostics for targeted lung cancer therapy. Theranostics 2022, 12, (9), 4147-4162.
  2. Kang, H.; Stiles, W. R.; Baek, Y.; Nomura, S.; Bao, K.; Hu, S.; Park, G. K.; Jo, M. J.; I, H.; Coll, J. L.; Rubin, B. P.; Choi, H. S., Renal clearable theranostic nanoplatforms for gastrointestinal stromal tumors. Adv Mater 2020, 32, (6), e1905899.
  3. Park, G. K.; Lee, J. H.; Soriano, E.; Choi, M.; Bao, K.; Katagiri, W.; Kim, D. Y.; Paik, J. H.; Yun, S. H.; Frangioni, J. V.; Clancy, T. E.; Kashiwagi, S.; Henary, M.; Choi, H. S., Rapid and selective targeting of heterogeneous pancreatic neuroendocrine tumors. iScience 2020, 23, (4), 101006.

To explain this more clearly, we added the following sentences in the Discussion section, lines 266-270: However, we previously demonstrated the difficulty of delivering drug molecules (> 300 Da) into panNET tissue due to the high tumor interstitial pressure [32]. Therefore, we decided to design a new delivery system for a quick-release of small molecule anticancer drugs to achieve sufficient treatment of panNET with reduced adverse effects.

  1. The authors suggest that encapsulating carboplatin reduces its cardiotoxicity, how? What is the hypothesis behind this, as long as the H-dots are not being targeted to a particular cell type using specific ligand-receptors, they can escape in the circulation.

The reviewer pointed out the most critical point of this study, and we apologize for not delivering this message clearly. The majority of panNETs are non-functional and do not express cell-type specific hormones, and thus, targeting panNETs has been a challenge. We hypothesized that most adverse effects of carboplatin including cardiotoxicity is resulted from the nonspecific distribution of drug to the off-target tissues and organs. Thus, by using nonsticky and noncytotoxic H-dot (Fig. 2), we might be able to reduce nonspecific background uptake in the surrounding tissues. Once H-dots meet the target tissue, the encapsulated carboplatin releases from the b-CD cavity quickly due to the hydrophilicity.

Although the mechanism of action is still unclear, H-dots were able to reduce the major adverse effect of carboplatin in major organs (Fig. 5). Further studies are required to clarify the mechanisms. We added this information in the discussion section to the lines 275-295.

  1. What was the reason to choose, carboplatin for this specific study? There are other safer drugs available which can be made more effective using combination strategies.

This is another critical point that we had to address in advance. We initially considered to utilize major platinum drugs including cisplatin, carboplatin, and oxaliplatin because they have been approved for clinical usage for various types of cancer treatments. However, cisplatin and oxaliplatin cause renal toxicity by damaging the kidneys and glomerular filters, which could be an issue since H-dots are exclusively excreted by the kidneys. On the other hand, carboplatin shows systemic toxicity including myelosuppression and cardiotoxicity, with rare cases of nephrotoxicity. In addition, the relatively high water solubility of carboplatin (14 mg/ml) is suitable compared with cisplatin (2.5 mg/ml) and oxaliplatin (4-8 mg/ml) to achieve a quick release from the H-Dot complexes before urinal excretion. We added a few sentences to the main text to explain this more clearly (Lines 54-59: “Here, we chose carboplatin as the encapsulated Pt drug for H-dot complexation since it shows systemic toxicity including myelosuppression and cardiotoxicity with rare cases of nephrotoxicity, while cisplatin and oxaliplatin cause renal toxicity by damaging the kidneys [7-9]. In addition, the water solubility of carboplatin (14 mg/mL) may be beneficial to achieve quick release from the H-dot complexes before urinal excretion.”

  1. In figure 1(c), if possible, the authors should use a white background.

We replaced the background color of color image in Figure 1(C) as suggested by the reviewer.

  1. Why was the PBS in drug release study, set at pH 6.0 and not 7.2?

We apologize for the typo This should read “pH 7.4” and we revised it accordingly.

  1. How is a early release of car from the nanocomplexes different from direct intake of car? Why was not a sustainable release over a period of time favourable for car delivery?

We appreciate the reviewer for this important question. As explained in the introduction, we tried to reduce the off-target toxicity of anticancer drugs by encapsulation them in H-dots. However, the short blood half-life and rapid renal clearance of H-dot, about 80% of injected dose within 4 h post-injection [20,21], have been a challenge. Therefore, we designed the drug/H-dot complex to release the encapsulated carboplatin quickly before they get excreted by the kidneys. To describe it clearly, we added the following sentence to Discussion.

Lines 282-285: Since one of the novel functions of the H-dot system is rapid renal clearance to reduce off-target effects of encapsulated drugs and carrier itself, the quick release of Car before urinary excretion would be critical for panNET treatment.

  1. Did the authors perform in vitro cytotoxicity at 48 hours, 72 hours? What was the difference observed?

Unfortunately, we didn’t perform the in vitro cytotoxicity assay over 24 h. Since the release profile of carboplatin from Car/H-dot was found to be very fast, we focused on the early time point of the toxicity assay in this manuscript. We appreciate the reviewer’s comment though.

Reviewer 2 Report

In this study, the authors demonstrated a delivery strategy of the chemotherapy carboplatin (Car), recognized for its relatively lower toxicity profile compared to cisplatin, to mitigate its off-target toxicity by utilizing the potential of the zwitterionic nanocarrier, H-dot. The authors confirmed that Car/H-dot nanocomplexes could target tumor surrounding areas by the enhanced permeability and retention (EPR) effect and showed sufficient treatment effect of Car/H-dot for in vitro and in vivo orthotopic panNET models. As almost 100% of Car release from the H-dot complex was observed within 24 h, the efficient release profile of Car/H-dot achieved the sufficient therapeutic effect of Car, which was accompanied by quick tumor targeting. In addition, Car/H-dot could prevent the loss of cardiac myofibrils as well as improve the blood AST level. Importantly, Car/H-dot nanocomplexes prevent the adverse effects of Car without reducting the treatment efficacy for insulinoma. Thus, the complex formation between Car and H-dot showed beneficial functions for panNET insulinoma treatment. These results may help further optimization of cancer treatment by H-dot complexes with several types of cancer therapeutics.

This is an interesting, detailed, comprehensive and important research that is seemingly (see comments below) demonstrating impressive results of diminishing Car systemic toxicity, through rapid renal clearance, while still exhibiting high anti-cancer capacity; even better than Car alone. Treatment of panNET using cisplatin (Cis) has been approved by the US Food and Drug Administration (FDA), but not yet treatment with Car. The authors suggest that the H-dot complex system described here, can improve therapeutic strategy for panNET by Car, which reduces the adverse effects compared with Cis.

However, there is one major criticism that really limit the value of these results; the size of the groups in all in vivo studies (Figures 3, 4. 5). All experiments are based on 3 mice/samples in a given treated group. See + s.e.m values in Figure 3e, Figure 4c,d for control and Car groups and the same for Fig. 5b,c.  Variability of in vivo studies is huge and an accepted size of a group in experimental biology is at least 5 mice/samples.

Authors should repeat the in vivo studies, at least the one described in Figure 5, with 5-7 mice per experimental group to validate the results.

Other comments:

- The design of the in vivo studies is completely not clear. It is written: “The insulinoma mice were divided into four treatment groups (n = 3) at 12 and 14 weeks old. Each group received different treatments: saline, 30 mg/kg Carboplatin, and 185 mg/kg Car/H-dot, given every 3 days 5 times within the 12-day”; there are only 3 experimental groups…what is the fourth? Were three group at the age of 12 weeks and the fourth at 14 weeks?? What does the fourth group of mice received as a treatment??? The design of the experiment should be clarified and accurately detailed.

- All reagents were injected intraperitoneally-why? What is the idea behind this approach when taking into consideration future human studies? Were i.v injections also tested?

- Results and discussion are written together and thus are hard to follow; highly suggested to be separated.

- H-dot nano-carriers should be a bit better and in more details described. Although being, most probably, the main research team dealing H-dot/chemotherapy complexes; the authors should describe whether more studies were performed dealing with such approach and complexes. In addition, authors should describe advantages as well disadvantages of using these nano-carriers mainly in regard with other widely used nano-carriers.  

Author Response

In this study, the authors demonstrated a delivery strategy of the chemotherapy carboplatin (Car), recognized for its relatively lower toxicity profile compared to cisplatin, to mitigate its off-target toxicity by utilizing the potential of the zwitterionic nanocarrier, Hdot. The authors confirmed that Car/H-dot nanocomplexes could target tumor surrounding areas by the enhanced permeability and retention (EPR) effect and showed sufficient treatment effect of Car/H-dot for in vitro and in vivo orthotopic panNET models. As almost 100% of Car release from the H-dot complex was observed within 24 h, the efficient release profile of Car/H-dot achieved the sufficient therapeutic effect of Car, which was accompanied by quick tumor targeting. In addition, Car/H-dot could prevent the loss of cardiac myofibrils as well as improve the blood AST level. Importantly, Car/H-dot nanocomplexes prevent the adverse effects of Car without reducting the treatment efficacy for insulinoma. Thus, the complex formation between Car and H-dot showed beneficial functions for panNET insulinoma treatment. These results may help further optimization of cancer treatment by H-dot complexes with several types of cancer therapeutics

.

This is an interesting, detailed, comprehensive and important research that is seemingly (see comments below) demonstrating impressive results of diminishing Car systemic toxicity, through rapid renal clearance, while still exhibiting high anti-cancer capacity; even better than Car alone. Treatment of panNET using cisplatin (Cis) has been approved by the US Food and Drug Administration (FDA), but not yet treatment with Car. The authors suggest that the H-dot complex system described here, can improve therapeutic strategy for panNET by Car, which reduces the adverse effects compared with Cis.

  1. However, there is one major criticism that really limit the value of these results; the size of the groups in all in vivo studies (Figures 3, 4. 5). All experiments are based on 3 mice/samples in a given treated group. See + s.e.m values in Figure 3e, Figure 4c,d for control and Car groups and the same for Fig. 5b,c. Variability of in vivo studies is huge and an accepted size of a group in experimental biology is at least 5 mice/samples. Authors should repeat the in vivo studies, at least the one described in Figure 5, with 5-7 mice per experimental group to validate the results.

We appreciate the reviewer’s constructive suggestions. Although we 100% agree with the reviewer, our lab is currently limited with further experiments of using insulinoma-bearing NOD/ShiLt-Tg(RipTAg)1Lt/J mice. We obtained this mouse line from the Jackson Laboratories (Bar Harbor, ME) a few years ago and actively maintained the colony without problems. However, recently we lost the final breeders and pubs, which caused discontinuation of this panNET mouse. We are currently actively negotiating with JAX to recover this line, but it will take time since we need to restart the process from cryo recovery. We will continue to evaluate the therapeutic efficacy of H-dot/drug complexes when we have more available animals. To answer the reviewer’s question, however, we collected all the previous data, and we were able to add preliminary experimental data, and we could increase the mice number to 4 mice/samples. This information is added to figure captions accordingly. We appreciate the reviewer’s kind understanding.

Other comments:

  1. The design of the in vivo studies is completely not clear. It is written: “The insulinoma mice were divided into four treatment groups (n = 3) at 12 and 14 weeks old. Each group received different treatments: saline, 30 mg/kg Carboplatin, and 185 mg/kg Car/H-dot, given every 3 days 5 times within the 12-day”; there are only 3 experimental groups…what is the fourth? Were three group at the age of 12 weeks and the fourth at 14 weeks?? What does the fourth group of mice received as a treatment??? The design of the experiment should be clarified and accurately detailed.

We apologize for the confusion, and revised the experimental design according to the reviewer’s comments.

Line 193: In this study, we injected 30 mg/kg of Car and 185 mg/kg of Car/H-dot, including the same dose of Car, into insulinoma mice (12-14 weeks old).

Line 428-430: The insulinoma mice were divided into three treatment groups (n = 3-4) between 12 and 14 weeks old. Each group received different treatments: saline, 30 mg/kg Carboplatin, and 185 mg/kg Car/H-dot, given every 3 days 5 times within the 12-day.

  1. All reagents were injected intraperitoneally-why? What is the idea behind this approach when taking into consideration future human studies? Were i.v injections also tested?

This is a very important question. Since platinum-based anticancer drugs are mostly injected intravenously in clinical situations, we also considered injecting H-dot complexes systemically. However, the majority of animal experiments published in the past show intraperitoneal (IP) administration of platinum-based anticancer drugs, although the reasons are not clearly described. Generally, IP injection is widely used for animal treatment because it is technically easy and suitable for repeated administration with less stress for animals. In addition, the IP administration of drugs in experimental animals has been reviewed as a justifiable route for pharmacological studies [Abdullah, A.S.; Sabrina, R.A.; Vardan, T.K., Intraperitoneal route of drug administration: Should it be used in experimental animal studies? Pharm Res 2020, 37, 12]. After IP injection, absorption of the small molecules (< 20,000 Da) mainly occurs by either diffusion or convection through the peritoneum into the blood capillaries, while molecules with higher than 30,000 Da enter the systemic circulation from the peritoneal cavity primarily via lymphatic vessels. Since the molecular weights of Car/H-dot (17,900 Da) and Car (371 Da) are smaller than 20 kDa, both molecules may follow the diffusion or convection model. However, it is also known that the difference in absorption models minimally affects the overall distribution of pharmacological agents from the peritoneal cavity to the systemic circulation. Therefore, we decided to inject H-dot complexes IP instead of IV. We appreciate the reviewer’s helpful comments.    

  1. Results and discussion are written together and thus are hard to follow; highly suggested to be separated.

We agreed with the reviewer and separated them in the revised manuscript according to the reviewer’s suggestion.

  1. H-dot nano-carriers should be a bit better and in more details described. Although being, most probably, the main research team dealing H-dot/chemotherapy complexes; the authors should describe whether more studies were performed dealing with such approach and complexes. In addition, authors should describe advantages as well disadvantages of using these nano-carriers mainly in regard with other widely used nano-carriers.

We appreciate the reviewer for constructive suggestions. We revised the main text as follows:

Lines 28-61: Given the substantial therapeutic impact of first-line Pt drugs on tumor tissues, diverse strategies have emerged to mitigate harm to normal tissues. These strategies encompass encapsulation within liposomes or polymeric micelles [10-12] and nanoparticle-based drug delivery [13,14]. These approaches augment the tumor targetability of encapsulated drugs, facilitated by the enhanced permeability and retention (EPR) effect. The EPR effect, acknowledged as a universal pathophysiological trait of solid tumors, fundamentally underpins the conceptualization and development of tumor-targeted delivery of anticancer drugs [15-17]. Previously, Kang et al. unveiled the intriguing phenomenon of the size-dependent EPR effect [18], where smaller polyethylene glycol (PEG) entities ( £20 kDa) exhibit a remarkable proclivity for tumor targeting while showing minimal nonspecific uptake [18]. Harvard-dots (H-dots) have been designed to deliver hydrophobic anticancer drugs, including imatinib, gefitinib, and genistein, with high delivery efficiency via the EPR effect relying on small-sized nanotherapeutics [19-21]. H-dots also offer reducing potential off-target toxicities of anticancer drugs through rapid renal clearance, of which the hydrodynamic diameter is smaller than the glomerular filtration threshold, approximately 5.5 nm [19]. Here, we chose carboplatin as the encapsulated Pt drug for H-dot complexation since it shows systemic toxicity including myelosuppression and cardiotoxicity with rare cases of nephrotoxicity, while cisplatin and oxaliplatin cause renal toxicity by damaging the kidneys [7-9]. In addition, the water solubility of carboplatin (14 mg/mL) may be beneficial to achieve quick release from the H-dot complexes before urinal excretion. We demonstrate the delivery of carboplatin using the EPR effect of H-dot by forming an inclusion complex to manifest enhanced therapeutic efficacy while mitigating the adverse effects through rapid renal clearance. We simply introduced other nano-carriers and our previous H-dot systems for tumor treatments in the Introduction section (Lines 39-43). To describe it more clearly, we added some of the below explanations in the Discussion section.

Lines 258-269: The treatment of panNET and avoiding normal pancreas represent a significant unmet clinical need. Intraoperative visual inspection and palpation performed by the surgeon is considered the most reliable source for panNET detection, which often results in incomplete resection of the tumor, with a 5-year disease recurrence rate of 10% and 97% of patients without and with liver metastasis, respectively [45-48]. Thus, complete tumor treatment is the only option for improving patient survival [48,49]. However, we previously demonstrated the difficulty of delivering drug molecules (> 300 Da) into panNET tissue due to the high tumor interstitial pressure [32]. Therefore, we decided to design a new delivery system for a quick release of small molecule anticancer drugs to achieve sufficient treatment of panNET with reduced adverse effects. Many drug carriers have been widely accepted for various kinds of tumor treatment for delivering the drugs into tumor tissue [10-14].

Line 276-296: We hypothesized that H-dot drug delivery systems might be able to achieve efficient delivery of chemotherapeutics with fewer adverse effects mainly due to the reduced off-target delivery and rapid renal clearance. Indeed, the drug/H-dot system showed high efficacy of delivering gefitinib and genistein to the lung cancers [20] and imatinib to the gastrointestinal tumors [19,21]. In this study, we successfully developed a quick-release system of Car from H-dot due to the hydrophilic property of drug. Over 90% of encapsulated Car was released from the Car/H-dot nanocomplex 8 h post-incubation. Since one of the novel functions of the H-dot system is rapid renal clearance to reduce off-target effects of encapsulated drugs and carrier itself, the quick release of Car before urinary excretion would be critical for panNET treatment. Indeed, sufficient treatment of panNET was observed both in vitro and in vivo, which did not disturb the efficacy of Car alone but rather slightly enhanced the in vivo treatment effects. The outstanding finding is that a quick drug release system of Car/H-dot could eliminate the panNET completely without drug-carriers penetration into the tumor tissue. While the limited penetration of drug delivery systems into deep tumor tissue has been one of the major limitations of nanoplatforms in treating panNET insulinoma [32], efficient Car release from the Car/H-dot complex around the tumor area was beneficial to the sufficient treatment. Furthermore, the H-dot delivery system prevented the cardiotoxicity of Car by encapsulation in the cavity of b-CD, which reduced cellular uptake (Fig. 2) and nonspecific uptake in organs (Fig. 5). Our results demonstrated Car/H-dot could eliminate panNET with fewer adverse effects. However, further investigations are required to clarify the mechanisms.

Reviewer 3 Report

The manuscript entitled “H-dot Mediated Nanotherapeutics Mitigate Systemic Toxicity  of Platinum-Based Anticancer Drugs” presents the preparation as well as in vitro and in vivo study of Car/H-dot nanocomplexes. Platinum-based anticancer agents have been widely used in the clinic to treat several types of cancers. However, serious undesirable side effects and intrinsic or acquired resistance limit their successful clinic use. The presented research suggests that Car/H-dot nanocomplexes have potential applications in cancer therapy. The authors managed to implement a combination of different in vivo and in vitro study and showed beneficial functions of Car/H-dot complex for panNET insulinoma treatment.

The paper is well written and structured. The design of the study is appropriate and cover a large number of parameters increasing the relevance of the study. The methods are well described and conclusions are clear and consistent with the results obtained.

I believe that the article can be published as is, given that the MDPI editors pay attention to typographical errors.

Minor editing of English language required.

Author Response

The manuscript entitled “H-dot Mediated Nanotherapeutics Mitigate Systemic Toxicity of Platinum-Based Anticancer Drugs” presents the preparation as well as in vitro and in vivo study of Car/H-dot nanocomplexes. Platinum-based anticancer agents have been widely used in the clinic to treat several types of cancers. However, serious undesirable side effects and intrinsic or acquired resistance limit their successful clinic use. The presented research suggests that Car/H-dot nanocomplexes have potential applications in cancer therapy. The authors managed to implement a combination of different in vivo and in vitro study and showed beneficial functions of Car/H-dot complex for panNET insulinoma treatment. The paper is well written and structured. The design of the study is appropriate and cover a large number of parameters increasing the relevance of the study. The methods are well described and conclusions are clear and consistent with the results obtained.

  1. I believe that the article can be published as is, given that the MDPI editors pay attention to typographical errors. Minor editing of English language required.

We appreciate the reviewer’s positive feedback. We revised typos and grammatical errors.

Round 2

Reviewer 2 Report

The authors accepted most of the comments and corrected the manuscript accordingly.